# Loss of Extrasynaptic Inhibitory Glycine Receptors in the Hippocampus of an AD Mouse Model Is Restored by Treatment with Artesunate

**DOI:** 10.3390/ijms24054623

**Published:** 2023-02-27

**Authors:** Jochen Kuhse, Femke Groeneweg, Stefan Kins, Karin Gorgas, Ralph Nawrotzki, Joachim Kirsch, Eva Kiss

**Affiliations:** 1Institute of Anatomy and Cell Biology, University of Heidelberg, 69117 Heidelberg, Germany; 2Institute of Neuroanatomy, Medical Faculty Mannheim, University Heidelberg, 68167 Mannheim, Germany; 3Department of Human Biology and Human Genetics, University of Kaiserslautern-Landau, 67663 Kaiserslautern, Germany; 4Department of Cellular and Molecular Biology, George Emil Palade University of Medicine, Pharmacy, Science and Technology of Târgu Mureș, 540142 Târgu Mures, Romania

**Keywords:** artemisinins, Alzheimer’s disease, glycine receptor, APP/PS1

## Abstract

Alzheimer’s disease (AD) is characterized by synaptic failure and neuronal loss. Recently, we demonstrated that artemisinins restored the levels of key proteins of inhibitory GABAergic synapses in the hippocampus of APP/PS1 mice, a model of cerebral amyloidosis. In the present study, we analyzed the protein levels and subcellular localization of α2 and α3 subunits of GlyRs, indicated as the most abundant receptor subtypes in the mature hippocampus, in early and late stages of AD pathogenesis, and upon treatment with two different doses of artesunate (ARS). Immunofluorescence microscopy and Western blot analysis demonstrated that the protein levels of both α2 and α3 GlyRs are considerably reduced in the CA1 and the dentate gyrus of 12-month-old APP/PS1 mice when compared to WT mice. Notably, treatment with low-dose ARS affected GlyR expression in a subunit-specific way; the protein levels of α3 GlyR subunits were rescued to about WT levels, whereas that of α2 GlyRs were not affected significantly. Moreover, double labeling with a presynaptic marker indicated that the changes in GlyR α3 expression levels primarily involve extracellular GlyRs. Correspondingly, low concentrations of artesunate (≤1 µM) also increased the extrasynaptic GlyR cluster density in hAPPswe-transfected primary hippocampal neurons, whereas the number of GlyR clusters overlapping presynaptic VIAAT immunoreactivities remained unchanged. Thus, here we provide evidence that the protein levels and subcellular localization of α2 and α3 subunits of GlyRs show regional and temporal alterations in the hippocampus of APP/PS1 mice that can be modulated by the application of artesunate.

## 1. Introduction

The strychnine-sensitive glycine receptors (GlyRs) represent the major neurotransmitter receptors at the inhibitory synapses in the spinal cord and brainstem [1], whereas the γ-aminobutyric acid type-A receptors (GABA_A_Rs) are the prevailing type of inhibitory receptors in higher brain regions [2]. The postsynaptic GlyRs are pentameric proteins comprised of alpha and beta-subunits, the latter serving, through binding to the scaffold protein gephyrin, the anchoring of the GlyR at postsynaptic membrane specializations [3,4]. Further, the incorporation of β subunits confers insensitivity towards the alkaloid picrotoxin, and induces an alteration of the glycine binding site at the interface of α- and β-subunits [5,6]. In addition to the heteromeric postsynaptic α/β GlyRs, different homomeric GlyRs are formed by alpha subunits that are mostly localized extrasynaptic and are involved in tonic inhibition [7,8]. Both subunits show significant sequence similarity to those of the nicotinic acetylcholine receptor (nAChR), GABA_A_R and serotonin type-3 receptor subunits, constituting a superfamily of ligand-gated ion channels (Cys-loop receptor family) [9]. Molecular heterogeneity of the GlyR results from the expression of four alpha- and one beta-subunit genes [2]. Alternative splicing and mRNA editing contribute further to GlyR-heterogeneity [10].

The subunit composition and distribution of GlyRs, including their subcellular localization, exhibit differential patterns depending on the region and developmental stage [11,12,13,14]. In contrast to the apparent wide distribution of GlyR β subunit transcripts in the central nervous system before and after birth, GlyRα mRNAs and proteins show a more restricted expression pattern. While α2 homomers represent the major GlyR isoform in the embryonic spinal cord, after birth the majority of the spinal cord GlyRs are formed by α1β and α3β heteromers, which are mostly localized at postsynaptic sites [14]. In the cerebral cortex and hippocampus, however, GlyRs are primarily composed of α2 or α3 subunits [15,16,17,18]. A detailed analysis of GlyR mRNA and protein expression in the rat hippocampus at different ages pointed to the predominance of extrasynaptic GlyRs in cell soma-rich layers, and many post-synaptic heteromeric GlyR α2β on dendrites at immature stages (P7). In the mature hippocampus (in 3-month-old rats), extrasynaptic α2/α3-containing GlyR were the most abundant [15] in concordance with earlier mRNA expression studies [11]. Moreover, in the retina, a close relationship was detected between the type of neuron and the prevalence of GlyR subunits expressed [19], suggesting a well-regulated recruitment of different GlyR subtypes for distinct physiological functions in different regions of the CNS.

Corresponding to their predominant extrasynaptic localization, hippocampal GlyRs are thought to play a tonic inhibitory role when activated by glycine, an important mechanism in controlling neuronal excitability [8,20]. Accordingly, exogenous administration of glycine suppressed neuronal hyperexcitability in the rat dentate gyrus in a strychnine-sensitive manner [21], and prevented somatically generated action potentials in both CA1 pyramidal cells and interneurons [22]. The complexity of glycinergic neurotransmission is further increased by the function of glycine as a co-agonist of the excitatory N-methyl D-aspartate receptors (NMDARs); thus, extracellular glycine levels modulate NMDAR-mediated responses and long-term potentiation (LTP) [13]. These observations strongly indicate that hippocampal glycine and its receptors may constitute an efficient system for maintaining an equilibrium between excitation and inhibition in hippocampal networks. This equilibrium might be affected in disorders of the CNS, possibly in a subtype, region and eventually disease stage-dependent manner.

The most prevailing form of dementia in humans is the sporadic, late onset Alzheimer’s disease (AD). AD is characterized by loss of synapses and neurons, which correlates with progressive loss of memory and cognitive capabilities [23,24]. Morphological hallmarks are extracellular deposits of β-amyloid protein (Aβ) in plaques and intracellular tangles composed of hyperphosphorylated tau protein [25], affecting primarily the hippocampus and cerebral cortex. Functional and structural changes of synapses in the AD brain comprise both the excitatory glutamatergic system, as well as the inhibitory GABAergic neurotransmission [23,24,25,26,27]. Several studies reported a robust change of the GABAergic system in AD patients and in multiple AD mouse models, including decreased levels of pre- and postsynaptic proteins. In particular, reduced levels and specific fragments of gephyrin were observed in post-mortem human AD brains [28,29]. Significant alterations in the structural and functional properties of GABA_A_ receptors were described in several AD models [30,31,32].

Previously, we reported a substantial reduction of the GABA_A_R-γ2 subunit and gephyrin protein levels in 12-month-old APP/PS1 mice, a widely used model of AD [31]. Interestingly, the reduction of these key proteins of inhibitory synapses in the brain could be restored by treating the APP/PS1 mice for three months with the anti-malarial drug artemisinin, or its semisynthetic derivative artesunate. In addition, we observed a clear reduction of amyloid plaque load with low-dose artemisinins. Accordingly, a dose-dependent decrease of soluble Aβ-concentrations and levels of C-terminal fragments (CTF) of the amyloid precursor protein (APP) were measured in hippocampus homogenates from artesunate-treated APP/PS1 mice [33].

Our recent findings, that artemisinins modulate the expression of GABAergic synapse proteins in APP/PS1 mice, and the general missing information concerning the glycinergic system in AD, prompted us to (1) analyze the protein levels and subcellular localization of GlyR α2 and α3 subunits as the most abundant subtypes in the mouse hippocampus in the early (3 months) and late stages (12 months) of AD pathogenesis, and (2) investigate whether these features are affected when APP/PS1 mice are treated with two different doses (10 mg/kg or 100 mg/kg) of artesunate (ARS), a derivative of artemisinin with higher oral bioavailability and better abilities to cross the blood-brain barrier than artemisinin does [34].

These studies, using immunofluorescence microscopy and Western blot analyses, revealed subunit-specific differences in the spatio-temporal and subcellular distribution of the GlyR α2- and α3 subunits in the CA1 region, and dentate gyrus of the hippocampus of 3-month-old compared to 12-month-old WT- and APP/PS1 mice. Interestingly, in 12-month-old APP/PS1 mice, protein levels of both α2 and α3 GlyR subunits were substantially reduced in the somatic layers of the hippocampus compared to WT mice. Moreover, artesunate treatment in these older APP/PS1 mice induced a recovery of GlyR protein expression in a subunit-specific and dose-dependent way. Finally, both a quantification of VIAAT-overlapping α2- and α3 GlyR clusters in the different hippocampus regions, as well as findings in the hAPPswe-expressing cultured hippocampal neurons treated with different concentrations of artesunate, strongly suggest that low doses of artesunate mainly target the extrasynaptic GlyR compartment.

## 2. Results

### 2.1. Protein Levels and Distribution of the α2 and α3 Subunits of GlyRs in the Hippocampus of 3-Month-Old Mice

In the present study, we analyzed the protein expression level of α2 and α3 subunits of GlyRs, as well as their regional distribution and subcellular localization in the hippocampus of APP/PS1 mice in the early stages (3 months) and late stages (12 months) of cerebral amyloidosis in comparison to WTs. Based on our recent findings, artemisinins can modulate expression level of GABAergic synapse proteins in APP/PS1 mice, inclusive of gephyrin, a scaffold protein common for GABAergic and glycinergic synapses. Thus, we also examined the hippocampus of APP/PS1 mice treated with two different doses of artesunate.

Coronal brain cryosections of WT and APP/PS1 mice (without and with treatment) were subjected to a detailed immunohistochemical analysis using GlyR α2 and α3 subtype-specific antibodies, applied previously also by others [7,35,36]. The pyramidal cell layer (PCL) and the stratum radiatum (SR) of the CA1 region and granular cell layer (GCL) of the dentate gyrus (DG) were systematically evaluated firstly in 3-month-old mice, an age when APP/PS1 mice present minimal or no amyloid plaques and cognitive impairment [37]. Our findings clearly show that both α2 and α3 subtypes of GlyRs are widely expressed in the different hippocampus subregions of 3-month-old WT mice (Figure 1A,B).

These results correlate well with earlier reports on mRNA and protein expression of GlyRs in younger rodents (9-week-old rats) [11,15], as well as with electrophysiological data from adolescent and young (P21–P50) C57/Bl6 mice [8]. Both anti-GlyR antibodies generated similar staining patterns and regional distribution (Figure 1A,B). Immunoreactivity was seen throughout the neuronal soma, as well as in dendrites in a highly dense punctate fashion, as known from earlier reports [15,16,17]. Regional variations were evident throughout the different hippocampal areas. Immunoreactivities were more prominent in layers containing the soma of pyramidal and granular cells than in the dendritic layers, indicating the higher density of perisomatic/somatic localized GlyR clusters—including a few hilar cells of the dentate gyrus (Figure 1A,B). Quantification of fluorescence intensities using confocal images (3 fields/section for CA1 and 2 fields/section for DG; *n* = 4–6 brains/group) revealed that the mean fluorescence intensities of both GlyR subtypes were 1.8 to 2.0-fold higher in the PCL and GCL than in the stratum radiatum representing the layer with fine dendritic processes of neurons (GlyR α2: 120.6 ± 37.5 and 106.3 ± 33.1 vs. 61.9; GlyRα3: 112.6 ± 13.7 and 101.9 ± 18.6 vs. 47.3 arbitrary units; Figure 1). We could not detect any evident differences, neither in staining intensities nor in the general regional distribution of the immunofluorescence signals for GlyR α2 and α3 subtypes in the hippocampus of 3-month-old APP/PS1 mice when compared to WT. Moreover, immunofluorescence signals were not evidently changed when the APP/PS1 mice were treated with artesunate for six weeks, independent of the dose applied (Figure 1A–D).

Earlier studies indicate that about two thirds of α2/α3 containing GlyRs in the hippocampus are localized extrasynaptic [15,16]. To prove these findings and detect possible differences induced by the APP/PS1 transgene in the subcellular localization of these GlyRs, we investigated brain sections double labeled for a GlyR subtype and the presynaptic protein VIAAT by confocal immunofluorescence analyses. VIAAT, the common neurotransmitter transporter of glycine and GABA, is located in both GABAergic and glycinergic synapses, and is widely used as a marker of all inhibitory pre-synaptic boutons [16]. To determine the numerical density of GlyR and VIAAT-colocalized clusters, indicating to some degree the synaptic localization of α2 and α3 subtypes, the overlay of GlyR clusters with VIAAT immunoreactivities was quantified in randomly selected areas of the PCL, SR of CA1 and GCL of the dentate gyrus (see Section 4 Materials and Methods). The results of these analyses supported the predominant extrasynaptic localization of both GlyR subtypes in the analyzed regions of the hippocampus in 3-month-old WT, and similarly in APP/PS1 mice, but with some differences in their regional distribution. We found that in the cell soma-rich layers (PCL, GCL) there were about 20% of the α2 clusters, whereas in the dendritic layer of CA1 (SR) only 13–15% of the α2 clusters were overlapping with VIAAT immunoreactivities (Figure 2), supporting somato-dendritic differences in the subcellular localization of α2 clusters (Figure 2C). The proportion of α3 immunoreactivities overlapping with VIAAT-positive clusters was about 10–15% in all layers analyzed (Figure 3). However, it should be noted that the precision of these values might be to some degree affected by the limitations in spatial resolution of the conventional light microscopy. Although the comparative analyses of 3-month-old APP/PS1 and WT mice revealed no statistically significant differences in the density of GlyR^+^/VIAAT^+^ clusters, there was a moderate decrease in the density of α2^+^/VIAAT^+^ puncta (Figure 2C). In parallel, a moderate increase in the density of α3+/VIAAT^+^ overlapping clusters in APP/PS1 mice in comparison to WT was consistently observed in all of the evaluated hippocampal subregions (Figure 3C). These findings may be of relevance, since APP/PS1 mice treated with low-dose artesunate seemed to prevent or reverse these changes., The quantification performed under similar conditions indicated a higher density of α2-, andin parallel, a lower density of α3 immunoreactivities overlapped with VIAAT-positive puncta in the pyramidal cell layer of CA1 of these treated mice in comparison to the untreated ones (Figure 2C and Figure 3C).

In summary, our analyses of 3-month-old mice demonstrate that in APP/PS1 mice, the protein levels of GlyR α2 and α3 subtypes are not changed compared to age-matched WT mice. However, subtle changes in the subcellular localization of these receptors can be observed, probably as part of the early remodeling of inhibitory neurotransmission, which might be prevented when the mice are treated with low-dose artesunate.

### 2.2. Protein Levels and Distribution of the α2 and α3 Subunits of GlyRs in the Hippocampus of 12 Month-Old Mice

Next, we examined the protein levels of GlyR subtypes in the hippocampus of 12-month-old APP/PS1 and age-matched WT mice (*n* = 4–6 animals/group). At this age, amyloid plaques and cognitive impairments can be clearly identified in APP/PS1 mice [37]. The labeling with the subtype-specific GlyR antibodies generated a similar distribution pattern, as shown in the 3-month-old animals. A rather intense immunoreactivity is evident in the somata of neurons of pyramidal (CA1) and granular cells of the dentate gyrus (Figure 4A,B). This is probably due to the high number of non-synaptic clusters in the cell soma, and is in accordance with the mainly extrasynaptic occurrence of α2/α3-containing GlyRs in the mature hippocampal neurons.

In the hippocampus of 12-month-old WT animals in comparison to 3-month-old mice, a remarkable and comparable increase of GlyR immunoreactivity could be observed within all hippocampus layers by ~60% for α2 (200.3 ± 12.78 vs. 120.6. 6 ± 37.5 in PCL, 189.4 ± 22.5 vs. 106.3 ± 33.1 in GCL, and 110.8 ± 16.8 vs. 61.9 in SR, arbitrary units), and by ~30% in soma-rich layers for α3 (149.6 ± 19.5 vs. 114.6 in GCL, 136.0 ± 26.4 vs. 101.9 in GCL) (Figure 1A,B and Figure 4A,B). Remarkably, GlyR α3 immunoreactivity remained unchanged in the dendritic layer of CA1 over time (47.6 ± 9.3 vs. 47.3 ± 8.3 in SR, arbitrary units). The same tendency was also detected for APP/PS1 mice (Figure 1A,B and Figure 4A,B).

In the hippocampus of 12-month-old APP/PS1 mice, in comparison to age-matched WTs, staining intensities of both α2 and α3 subtypes in all hippocampus regions analyzed were diminished (Figure 4A,B). Gray scale values measured in the different hippocampus layers confirmed a statistically significant reduction in mean fluorescence intensities of the GlyR α2 subtype in the PCL and GCL of CA1 (Figure 4C), and of α3 subtype in the PCL of CA1 (Figure 4D). The reduction of the GlyR α2 and α3 protein levels in 12-month-old APP/PS1 mice were sustained by the immunoblot data of total hippocampus homogenates (Figure 4E–H). The analyses of the hippocampi of 12-month-old APP/PS1 mice treated with low and high-dose artesunate, starting at 9 months of age, provided some unexpected results. The decreased levels of GlyR α3 within the soma-rich layers of the hippocampus observed in untreated APP/PS1 mice were restored with high consistency to about WT levels in mice treated with low-dose artesunate. This effect showed a higher variability by the high-dose treatment (Figure 4C,G). In contrast, the decreased GlyR α2 immunoreactivities were not significantly changed by the artesunate treatment. In some animals, the high-dose artesunate even led to a further decrease in α2 staining intensities (Figure 4D), a finding which was also supported by the WB data of total hippocampus homogenates (Figure 4G,H). Interestingly, when quantifying the colocalization of GlyR and VIAAT clusters, no obvious differences between experimental groups were found for the number of overlapping VIAAT and GlyRs immunoreactivities, neither for α2 nor for α3. There was an average coincidence for α2 of 30–40% in PCL, 12–15% in SR of CA1 and ~30% in GCL of the dentate gyrus (Figure 5A), and for α3 of 10–11% in PCL, 17–19% in SR of CA1 and 12–14% in GCL of the dentate gyrus (Figure 5B). One exception was the significantly higher density of α3^+^/VIAAT^+^ signals in all analyzed layers of the hippocampus in the high-dose ARS-treated APP/PS1 mice (Figure 5B), indicating dose-dependent differences in the effects of artemisinins, as observed in earlier studies [33]. It should be again noted that the precision of the absolute values provided by the quantification may be to some degree affected by resolution limits of the conventional light microscopy.

The comparison with 3-month-old mice revealed that the proportion of α2 GlyR subunits that overlap with VIAAT-positive immunoreactivities increased over time, being almost double at 12 months of age in both the PCL (30–40% vs. 18–20%) and GCL (~30% vs. 18–20%), since at the dendritic synapses of the stratum radiatum, a slight decrease was observed. The density of these putative synaptic α3 GlyR subunits was not evidently different between 3 and 12 months on somatic synapses in PCL and GCL (10–12% at both time points), whereas the density at the dendritic synapses in the SR of CA1 increased (17–19% vs. 12–13%) (Figure 2C, Figure 3C and Figure 5).

In summary, our analyses of 12-month-old mice demonstrate that in the hippocampus of APP/PS1 mice modeling late-stage AD, the protein levels of α2 and α3 subtypes of the GlyR were significantly decreased compared to age-matched WT mice. Despite the evident decrease in total protein levels, the proportion of GlyR subunits overlaying with VIAAT immunoreactivities (and thus, presumed to be synaptic localized) was not changed in the different hippocampus layers. Treatment with low-dose artesunate led to a complete rescue of protein levels of the GlyR α3 subunit, without affecting the density of these putative synaptic localized clusters. This indicates that the changes in expression levels primarily involve the extracellular localized GlyR subunits, and points towards a role of slow tonic inhibition by the activation of extrasynaptic α2/α3 GlyRs in the mature hippocampus.

### 2.3. Distribution of GlyRs in hAPPswe Expressing Cultured Hippocampal Neurons

To further prove and examine the effects of artesunate on GlyRs in conditions that mimic an AD environment, we treated cultured hippocampal neurons that expressed the human APPswe protein with increasing concentrations of artesunate (from 0.1 µM to 1 µM) for seven days. Previously, we have shown that this treatment paradigm reduced the generation of C-terminal fragments in hAPPswe-infected hippocampal cell cultures [33]. Hippocampal neurons transduced with a cMyc-tagged hAPPswe transcript to induce overexpression of mutated hAPP were immunolabelled with an anti-myc antibody (to detect transduced neurons), in combination with an antibody for GlyRs (α1, α2, α3, mAb4a) and the presynaptic marker VIAAT on div15 (Figure 6).

Cluster density and size were analyzed at both the soma and proximal dendrites (Figure 6B). In our hands, only the mAb4 antibody produced signals that allowed a reliable comparative analysis of cells under different experimental conditions. An antibody specific for α2-containing GlyRs also yielded clear clusters; however, here the interindividual variability within treatment groups was very high, possibly obscuring more subtle differences between groups (Appendix A). For α3, none of the commercially available antibodies was able to evidence specific clusters in the dissociated hippocampal neurons. Thus, we focused our analyses of GlyRs detected by the mAb4a antibody. Div15 hippocampal neurons showed clear mAb4a clusters and a more diffuse cytoplasmic staining. Only a moderate number of the mAb4a clusters overlapped with presynaptic VIAAT signals. Clusters overlapping with VIAAT were, on average, 20% bigger than their extrasynaptic counterparts in untreated control cells (putative synaptic clusters 0.061 and 0.074 µm^2^, and extrasynaptic clusters 0.051 and 0.060 µm^2^ at the soma and proximal dendrites, respectively). Overexpression of hAPPswe did not change either the density or the size of mAb4a clusters (Figure 6B,C). Next, we investigated the effect of artesunate treatment on the same parameters within the hAPPswe overexpressing neurons. Here, we found a clear concentration-dependent increase in the number of GlyR clusters at both dendritic and somatic compartments after treatment with artesunate (Figure 6B). The effect was most pronounced with 0.25 µM and 1 µM in the somatic compartment, and with 0.25 µM and 0.5 µM artesunate in the dendritic compartment (Figure 6B,C). In the somatic compartment, the same artesunate dosages also led to a subtle but statistically significant increase in cluster size (Figure 6C). Interestingly, the increase in cluster density was observed only for extrasynaptic clusters, and the density of putative synaptic clusters remained unchanged.

In summary, our in vitro experiments with hAPPswe-transfected hippocampal neurons provide additional evidence that artesunate in low concentrations can increase the density of GlyRs, and that this mainly involves extrasynaptic GlyRs, which is in line with our in vivo findings in 12-month-old APP/PS1 mice.

## 3. Discussion

In the present study, we demonstrate that artesunate—a derivative of the herb plant-derived artemisinin—modulates the protein levels of GlyR subunits in conditions of increased amyloidogenesis, both in vivo and in vitro experiments in an age and a concentration-dependent manner. In particular, in the hippocampus of double transgenic APP/PS1 mice, GlyR α3 subunit levels—found to be significantly reduced during late stages of cerebral amyloidogenesis—were restored after treatment with a low-dose (10 mg/kg) artesunate. In cultured hippocampal neurons transfected with hAPPswe treatment with artesunate at a concentration as low as 0.25 μM, already significantly increased GlyR protein levels and GlyR clustering. In addition, our findings strongly indicate that low-dose artesunate mainly targets the extrasynaptic GlyR compartment. This conclusion is supported by the quantification of α2- and α3 GlyR clusters, overlapping VIAAT immunoreactivities in the different regions of the mouse hippocampus and in hAPPswe expressing cultured hippocampal neurons treated with different concentrations of artesunate.

Artemisinins have been used for a long time in traditional Chinese medicine [38,39]. Today, artemisinin and its derivatives are applied in combination therapy for the treatment of malaria [40]. Moreover, these drugs exert anti-oxidant, anti-inflammatory, anti-carcinogenic and anti-viral effects [41,42,43,44]. Proposed mechanisms of actions include the modulation of oxidative stress caused by mitochondrial ROS production and imbalance of antioxidant enzymes, protection of the mitochondrial membrane potential required for ATP production, and interference with various anti-inflammatory and anti-apoptotic signaling pathways, such as NF-kappaB and PI3 kinase/Akt, and ERK/CREB/Bcl-2 and Nrf2 [38,45,46]. Recently, it was also shown that artemisinins improve cognitive capacities in AD-models reducing Aβ levels, inflammation and amyloid plaque load, and were recommended as potential candidates for the treatment of Alzheimer’s disease in humans [45,46,47,48,49]. In agreement with these data, our recent study analyzing the effects of artemisinin and artesunate in APP/PS1 mice confirmed their potent anti-amyloidogenic properties, and identified for artesunate the limitation of APP-processing as one effective mechanism for mediating these effects, especially when used in a lower dose [33].

Simultaneously, other in vitro studies provided evidence that artemisinins also affect the inhibitory neurotransmission, though with somehow controversial results. On the one hand, it was demonstrated that artemisinins reduce GABAergic signaling by inhibition of GABA synthesis and GABA_A_ receptor anchoring at the postsynaptic scaffold protein gephyrin [50,51,52], with the latter relying on the ability of artemisinins to directly target a binding pocket of gephyrin responsible for receptor-anchoring [50,53]. In studies with a primary cultured spinal cord and hippocampal neurons, these authors also described a compound- as well as time-dependent regulation of inhibitory glycinergic currents and gephyrin-GABA_A_R co-clustering by artemisinins. For example, treatment with 10 μM of artesunate reduced glycinergic currents, and 1h treatment with 50 μM of artesunate led to a significant decrease in cluster number, area, and fluorescence intensity. In contrast, pretreatment for 24 h had no relevant effect on receptor clustering, with a longer exposure to these ARS concentrations being even toxic. On the other hand, in our own in vivo studies, artemisinins restored the altered expression of key proteins of inhibitory synapses in an AD mouse model [33]. Moreover, in cultured pancreatic islet cells, treatment with 10 μM artemether for three days enhanced GABA-ergic signaling in a gephyrin-dependent manner and raised insulin secretion [54]. Interestingly, a recent study revealed that the treatment of zebrafish with artesunate at relatively high doses induced acute cardiotoxicity, whereas a low-dose treatment exerted cardioprotective effects [55]. These results further suggest that the effects of artesunate may be dose- and state-dependent. Thus, artemisinins seemingly display the specific hormetic behavior of potentially toxic substances that may provide beneficial functions when administered in low doses [55].

In the present study we analyzed the protein levels, regional distribution and subcellular localization of α2 and α3 GlyR subtypes by immunofluorescence microscopy and Western blotting. The staining pattern and distribution generated by the subtype-specific antibodies used in our experiments corresponded to those described for GlyRs in earlier immunohistochemical studies in rodents, and support the wide presence of GlyRα2 and GlyRα3 subtypes in all subregions of the young and adult mouse hippocampus [15,16,17].

The overall distribution pattern showed a higher expression within the stratum pyramidale of CA1 and stratum granulosum of DG, and indicated the predominant somatic localization of these GlyRs. This overall pattern was not evidently different in APP/PS1 mice in comparison to WTs, neither at three months nor at 12 months of age. In contrast to the repeatedly detected increased levels of gephyrin and γ2 subtype of GABA_A_R in the hippocampus of 3-month-old APP/PS1 mice in comparison to WTs, and a further increase after treatment with artemisinin [31,32,56,57], we could not detect any significant differences in the region-specific protein levels of GlyRα2 and GlyRα3 between different experimental groups, suggesting different regulatory mechanisms for GABAergic and glycinergic inhibition in early stage AD. Indeed, by co-localization studies with presynaptic VIAAT, we detected significant but opposite changes in the subcellular localization of GlyR subtypes in the hippocampus of these young APP/PS1 mice. Specifically, low-dose artesunate treatment increased the number of VIAAT overlapping α2 clusters, mainly in the pyramidal cell layer of the CA1 region, whereas the density of α3 clusters overlapping with VIAAT was significantly reduced in this layer of the hippocampus. Thus, low-dose artesunate not only prevented the opposite, although moderate alterations in the subcellular localization of α2 and α3 GlyR clusters observed in APP/PS1 mice in comparison to WT, which might be part of the early remodeling process of inhibitory neurotransmission in AD, but even corrected these parameters over WT levels.

At 12 months of age in the hippocampus of APP/PS1 mice, the protein levels of GlyR α2 and α3 subunits were decreased compared to age-matched WT mice, most evidently in cell soma-rich layers of the hippocampus, where in general GlyRs are most abundant. These might be part of the typical loss of synaptic proteins in the brain occurring at later stages of AD, since previously we detected a significant decrease of other inhibitory synaptic components including gephyrin and GABA_A_R-γ2, and to some extent VIAAT in the hippocampus of 12-month-old APP/PS1 mice. Our findings correlate to those reported in 6–7-month-old 2 × Tg mice (another double transgenic model of AD), showing a reduced level of GlyR expression in the nucleus accumbens (nAc) parallel to a reduction in the pre- and post-synaptic markers SV2 and gephyrin. Moreover, these authors demonstrated a concomitant increase in intraneuronal Aβ accumulation, as well as occasional extracellular amyloid deposits in nAc [58]. Additionally, a decrease in glycinergic miniature synaptic currents in nAc-brain slices was recorded, likely reflecting a decrease in the number of receptors in the AD-altered neurons. Interestingly, it was also found that the function of synaptic receptors was not affected, suggesting that extrasynaptic receptors might be compromised. Our results in the 12-month-old APP/PS1 mice are in line with these data, since despite an evident decrease in total protein levels, the proportion of putative synaptic localized receptor subunits in the analyzed layers of CA1 and DG were not altered in comparison to WTs, altogether indicating that the changes in the expression levels of GlyRs in AD should primarily involve the extrasynaptic localized GlyR subunits.

Amyloid β (Aβ) is considered to be one of the main pathological agents in AD and induces a plethora of synaptic deficits, including downregulation of post-synaptic receptors and neurotransmitter release [25]. Interestingly, it was demonstrated that Aβ protein (1–42) interacts with GlyRs in rat hippocampal pyramidal cells, probably by binding to external sites on GlyRs, and participates in the modulation of tonic inhibition by these receptors. Specifically, it was shown that relatively high (100 nM) Aβ concentrations efficiently reduced the current through GlyRs [59], thus suggesting that the suppression of the tonically active glycine receptor by Aβ in the AD brain may result in increased excitability of the hippocampal neurons.

The functional impact of GlyRs in the hippocampus has gained attention only in recent years. It is well-established that these receptors mediate fast inhibitory synaptic transmission in the spinal cord and brainstem, and are involved in motor coordination (locomotor behavior) and the processing of pain perception [1,2]. Less is known about their functional role in higher brain regions. In the hippocampus, GlyRs are thought to modulate the excitability of neuronal networks, cellular homeostasis and cell viability [13,60,61] and thus underlie functions such as learning and memory. Correspondingly, silencing the gene coding for the GlyR α2 subunit in mice resulted in impaired adult hippocampal neurogenesis associated with deficits in spatial memory, when tested in the Morris water maze [62]. Abundant evidence supports a contribution to tonic hippocampal inhibition and indicates an anticonvulsive potential of GlyR activation [13,21]. Indeed, inhibition of GlyR-currents in the hippocampus may lead to hyperexcitability and seizures [63]. Furthermore, it is well known that patients with AD and animal models are more prone to epilepsy than the population average [64]. Thus, one might suppose that the suppression of glycine receptor mediated tonic inhibition by Aβ may be one possible mechanism that links Aβ accumulation in AD to the development of epilepsy.

In particular, extrasynaptic GlyR harboring α2 or α3 subunits are proposed to have an important function in mediating slow tonic GlyR activation in the mature hippocampus [8,60]. In agreement with previous reports [12,15,20], our findings support the predominant extrasynaptic localization of both α2 and α3 GlyRs subunits in young and adult mouse hippocampus. We found that non-synaptic localization is most prominent in the case of somatic localized α3 GlyRs, similar to earlier findings in the rat hippocampus [15], with α3 GlyRs showing only ~10% postsynaptic localization in comparison to ~30–40% of α2 GlyRs in the pyramidal cell layer of CA1, as well as in the granule cell layer of DG in 12-month-old mice. Interestingly, the treatment of adult APP/PS1 mice with artesunate clearly affected the expression of α3 GlyRs, whereas the expression level of α2 GlyRs was not significantly increased. The relative density of putative synaptic localized GlyR clusters was not affected by artesunate treatment at this age, suggesting that the changes in expression levels primarily involve the extracellular localized GlyR subunits. In this context, it is interesting to note that the increase of GlyR subunit expression in cultured hippocampal neurons after treatment with low artesunate concentrations in our experiments relied primarily on the robust increase of extrasynaptic GlyR clusters at both the somatic and the dendritic compartments. Altogether, these findings indicate that treatment with artesunate might support the slow tonic inhibition mediated by GlyRs in the hippocampus in such a way as to restore functionally relevant intrinsic homeostatic paradigms altered in AD pathogenesis. It is noteworthy that artemisinins were reported to effectively control epileptic seizures in AD-independent conditions [45], which indicates that the effects of artesunate on GlyRs, and in general on inhibitory neurotransmission in the AD brain, cannot simply be the result of their anti-amyloidogenic properties.

Overall, our present study suggests that, in addition to the GABAergic inhibitory system, GlyRs are also altered in the hippocampus of APP/PS1 mice, which in turn might contribute to AD pathogenesis—and thus should be considered in the design of future therapeutic approaches. Artesunate might be a potential candidate, however, considering the relative pronounced condition and dose-dependent effects of artemisinins, comprehensive evaluation and further studies are required.

However, the findings of this study have to be seen in light of some limitations.

The first limitation arises from the low spatial resolution of light microscopy used in this study to identify synaptic localized GlyRs in double immunolabeled brain sections. The limit of an optical microscope to view adjacent structural details clearly may result in the generation of pseudo clusters, and thus represents a confounding factor for synapse density quantification. Thus, we cannot rule out the possibility that some of the GlyR α2 or GlyR α3 positive puncta apparently overlapping VIAAT immunoreactivities observed in the mouse hippocampus sections are from extrasynaptic sites. Future studies using super-resolution imaging techniques, that are able to localize isolated fluorescent molecules with precision well beyond the diffraction limit, can serve to further validate and refine the calculation of absolute protein densities.

The second limitation concerns the antibodies used in this study for the detection of GlyR α2 and GlyR α3 subunits. Although validated for the methods used in our study, the specificity of these antibodies was not validated by knockout (KO) technology; thus, a cross-reactivity with other proteins e.g., other GlyR subunits cannot be completely excluded. Upcoming analyses using GlyR α2 and GlyR α3 KO cells or tissues have to ultimately clear the subunit specificity of these antibodies, and thus refine or reinforce the findings of this study.

Third, the experimental design of this study does not include WT mice treated with artesunate. Further studies analyzing the effects of artemisinins also on physiological conditions can provide additional information and help to better understand their mechanisms of actions in pathological conditions, such as Alzheimer’s disease.

## 4. Materials and Methods

### 4.1. Animals and Tissue Preparation

Male double transgenic APP/PS1 mice and age-matched nontransgenic control mice (WT) of the same genetic background (C57BL/6) were used for the experiments. APP/PS1 mice were obtained from Prof. Dr. M. Jucker (University of Tübingen, Germany) and bred in the animal care unit of the University of Kaiserslautern. Offspring were genotyped as described earlier [65].

APP/PS1 mice co-express the KM670/671NL “Swedish” mutated amyloid precursor protein (APP) and the L166P-mutation-carrying human presenilin 1 (PS1) under the control of a neuron-specific Thy1 promoter element. Due to the elevated secretion of human Aβ-peptide, the mouse model mimics aspects of cerebral amyloidosis. The first amyloid plaques appear in the dentate gyrus at 2–3 months of age, and in the CA1 region of the hippocampus at about 4–5 months of age [7]. Deficits in the Morris water maze test and impairments of LTP in the hippocampal sub-region CA1 were reported, starting at 7–8 months of age [65]. Thus, to model early and late stages in AD-pathogenesis, 3- and 12-months-old animals were used for analyses, respectively. For 3 months experiments on 6-week-old APP/PS1 mice, and for 12 months experiments on 9-month-old APP/PS1 mice were randomly assigned to three experimental groups: APP/PS1 (A) mice on chow diet, and two artesunate-treatment groups, artesunate (Cayman Chemical, Ann Arbor, MI, USA) being mixed in the chow rodent diet at a dose of 10 mg/kg and 100 mg/kg, respectively (A-ARS10 and A-ARS100), for a further 6 weeks or 3 months, respectively. The doses of artesunate used were based on published data after apprehensive literature research [34,38]. At the end of the experiments, the mice (4–6 animals/group) were deeply anesthetized with isoflurane and decapitated. Brain hemispheres were separated. For immunohistochemistry, one hemisphere of each brain was mounted in an OCT embedding compound (VWR Chemicals) and snap-frozen in an absolute ethanol-dry ice mixture. For biochemical analysis, the fresh hippocampus was dissected from the brain hemispheres and immediately frozen in liquid nitrogen. Tissue samples were stored at −80 °C until use [31].

All animal procedures were carried out in accordance with the European Communities Council Directive (86/609/EEC) and approved by the responsible regional state authorities (T-65/15 G-72/17 and (23 177-07/G 18-2-041).

### 4.2. Immunofluorescence and Confocal Microscopy of Tissue Sections

*Immunolabeling.* The protocol was described in detail previously [31,32]. Briefly, coronal cryostat sections (8 μm) cut from fresh-frozen brain hemispheres were fixed with 4% (*w*/*v*) PFA (Roth-Histofix 4%) for 8 min and incubated overnight at 4 °C with primary antibodies diluted in blocking solution without Triton X-100. Individual sections were incubated with a primary antibody against GlyRs separately, or in combination with a polyclonal guinea pig anti-VIAAT antibody (Synaptic Systems, Göttingen, Germany, cat. nr. 131308, 1:500). The primary antibodies for detection of GlyRs included: rabbit polyclonal anti-GlyR α2 (GeneTex, Irvine, CA, USA, GTX105634, 1:100), rabbit polyclonal anti-GlyR α3 (Sigma-Aldrich, St. Louis, MO, USA, AB15014, 1:200). For visualization of primary antibodies, sections were incubated with corresponding secondary antibodies conjugated to fluorophores (Vector Laboratories, Invitrogen, Jackson Immunoresearch Laboratories, Waltham, MA, USA) (60 min, RT), followed by incubation with DAPI (4′,6-diamidino-2-phenylindole, 1:200) to visualize nuclear DNA. Controls omitting the primary antibodies were included. Serial sections from untreated and artesunate-treated mice for each time point were labelled simultaneously, using the same batches of solutions to avoid differences through immunolabeling conditions.

*Confocal laser scanning microscopy and quantitative analysis.* Confocal microscopy was performed with a Leica TCS SP8 microscope (Leica Microsystems CMS GmbH, Mannheim, Germany) as described previously [27,29]. Briefly, the emission filter settings were 490–540 nm for PMT2 (green) and 630–665 nm for PMT4 (cyan). Using a HC PL APO CS2 63.0 × 1.40 oil objective, images were acquired in sequential mode with a frame average of 4. Stacks of 8 optical sections (1024 × 1024 pixels) spaced by 500 nm were recorded. Laser power and settings were identical for all samples stained in one session. Three randomly chosen fields within CA1 and two in the dentate gyrus (DG) of each hippocampus (*n* = 4–6 brains/group) were recorded for quantitative analysis. In each field, rectangular areas of 500 × 150 μm within the pyramidal cell layer (PCL) and the stratum radiatum (SR) of CA1 and the granular cell layer (GCL) of DG were randomly selected and quantified for mean fluorescence intensities using ImageJ/Fiji software (http://pacific.mpi-cbg.de/wiki/index.php/Fiji (accessed on 15 January 2020)). Mean fluorescence intensity of a region of interest was calculated from maximal intensity projection of eight optical sections. Mean values calculated for each animal were used for final statistics. WTs were set to 1, and data are given as mean ± SEM [31,33].

To quantify the colocalization of GlyRs with the presynaptic inhibitory protein (VIAAT+) in the PCL and SR of the CA1 area and GCL of the dentate gyrus, an ImageJ/Fiji macro developed in the Cell-Networks Math-Clinic Core Facility of the University of Heidelberg was used [32]. This first semi-automatically segmented the immunofluorescent puncta using the threshold method, and then automatically processed the generated binary masks to find overlapping signals between the different confocal channels. Each staining in a double labelling image (GlyR subunit and VIAAT) was subjected to the same intensity threshold cut-off, binary mask generation and particle analysis detection. The number and area of overlapping puncta were then counted by the ImageJ/Fiji macro, and a customized summary table was created in the output directory for each processed image for further validation and statistical analysis. The percentage of overlapping (synaptic) puncta was then calculated from the total number of puncta positive for the respective GlyR subunit [15,66]. Mean values calculated from 2–3 CA1 and 2 DG images for each animal were used for final statistics [32,56].

### 4.3. Protein Extracts and Immunoblot Analysis

Frozen hippocampus tissue samples were homogenized, as described elsewhere [31]. Proteins were blotted on polyvinylidene difluoride membranes (Millipore, Burlington, MA, USA) according to the manufacturer’s instructions, and incubated with the primary antibodies at 4 °C overnight. For detection of GlyRs, the same antibodies were used as for immunohistochemistry: rabbit polyclonal anti-GlyR α2 (GeneTex, 1:100), rabbit polyclonal anti-GlyR α3 (Sigma-Aldrich, 1:200) and a mouse anti-β-actin antibody (Sigma-Aldrich, A5441, 1:40,000). The corresponding horseradish peroxidase-conjugated secondary antibodies (Bio-Rad, Hercules, CA, USA) were detected using the ECL Prime detection kit (Amersham Biosciences, Amersham, UK). After exposure to hyperfilms (Amersham Bioscience), pixel intensities of the bands of interest were analyzed using ImageJ. Band intensities normalized to beta-actin were averaged from three to four animals (hippocampal tissue/group) and results are presented as mean ± SD.

### 4.4. Hippocampal Cell Cultures

*Construction of expression plasmids*: DNA fragments encoding the complete coding sequence of human APP695 wild-type and human APPswedish (APP695 KM670/671NL) mutant, which have been tagged with a N-terminal myc-epitope coding sequence cloned into the Kpnl site of an APP cDNA [67], Eggert S, unpublished] were amplified by PCR, creating BamH1 and EcoRI flanking restriction sites. These PCR-fragments were subsequently inserted into the BamHI and EcorRI restriction sites of the lentiviral vectors pFUGW [68] and pFSGW [69], replacing the corresponding EGFP encoding BamH1 and EcoRI fragments of these plasmids. An ubiquitin promotor (U) in the vector pFUGW or a human synapsin promotor (S) in pFSGW drove the expression of the described APP constructs.

*Lentivirus preparation*: Recombinant lentiviral particles were produced as described previously [70]. In brief, HEK293LTV cells (Cell Biolabs, San Diego, CA, USA) were transfected with equimolar amounts of pFUGW, pΔ8.9 and pVSVg using polyethyleneimine (Sigma-Aldrich). Virus particles were harvested two days after transfection by collecting the cell culture supernatant and concentrated by ultracentrifugation (90 min at 75,000× *g* (25,000 rpm, SW32Ti rotor, Beckman Coulter, Brea, CA, USA)). Virus containing solution was aliquoted, shock frozen in liquid nitrogen and stored at −80 °C for further use.

*Cell culture.* Primary cultures of rat hippocampal neurons were prepared from E19 embryos plated at a density of 40,000 cells/cm^2^ into 24-well plates. Infection with lentivirus dilutions was performed two days after plating (days in vitro/div2). Artesunate was diluted in DMSO and added to the cell culture medium in increasing concentrations (0.05, 0.125, 0.25 μM) at div 7–8. Cells were fixed for immunocytochemistry seven days after treatment, starting at div 14–15 [33,56].

*Immunolabeling.* The protocol was described in detail [67]. Briefly, coverslips containing cultured neurons were rinsed once with PBS and subsequently fixed with 4% (*w*/*v*) PFA for 10 min at RT. After washing, cells were incubated with preheated 10 mM sodium citrate with 0.05% Tween for 20 min at 95 °C for antigen retrieval, followed by incubation with blocking buffer (5% horse-serum, 1% BSA and 0.1% TritonX-100 (Roth, Karlsruhe, Germany) in PBS) for 30 min. Subsequently, cells were incubated with up to three primary antibodies overnight at 4 °C including anti-GlyR α2 (rabbit polyclonal, GeneTex, 1:100), mAb4a (mouse monoclonal [68]), anti cMyc-A488 (mouse monoclonal, 9E10, Abcam) and VGAT (guinea pig polyclonal, Synaptic Systems). The following day, samples were washed three times in PBS, whereafter they were incubated with appropriate secondary antibodies conjugated to fluorophores (Vector Laboratories, Invitrogen, Jackson Immunoresearch Laboratories) for 30 min at room temperature. Both primary and secondary antibodies were diluted in blocking solution without TritonX-100. After a further three washes in PBS, samples were rinsed in demineralized water and the coverslips were mounted onto glass slides using Mowiol/Dabco (Roth, Karlsruhe, Germany). The slides were kept at 4 °C in the dark until imaging.

*Confocal laser scanning microscopy and quantitative analysis.* Confocal microscopy was performed with a Leica TCS SP8 microscope (Leica Microsystems CMS GmbH, Mannheim, Germany). Transduced hippocampal dissociated neurons were visually identified with cMyc staining. Only cells with a clear cytoplasmic cMyc signal were selected for imaging. For quantification of GlyR clusters, cultures were stained with either mAb4 (all GlyR subunits) or an alpha2-specific antibody, in addition to cMyc and VIAAT as a marker of presynaptic inhibitory sites. For each cell, a 92 × 92 µm region containing the somatic compartment and proximal dendrite(s) was recorded as 4-channel, sequential Z-stack. All settings, including laser power, were kept constant between all recordings in one data set. Particle analysis was performed using NIH’s Fiji. In short, regions of interest (ROIs) containing all clusters within the most proximal part of the apical dendrite or of the soma were manually drawn on the Z-projections. Within the selected ROI, the GlyR or VIAAT signal was thresholded (thresholds were kept constant throughout all recordings) and all particles of at least 0.04 µm^2^ were measured for size, average and maximal grey intensities. VIAAT positivity was used to subdivide clusters as synaptic or extrasynaptic. Three different experiments with eight pyramidal neurons/experiment were analyzed, if not otherwise specified. Statistics were performed as one-way ANOVA, with *n* = number of neurons.

### 4.5. Statistical Analysis

Statistical evaluation was done in Prism (GraphPad Software Inc., San Diego, CA, USA). Statistical significance of IF and immunoblot data were determined using one-way ANOVA with Bonferroni’s multiple comparison test. Statistical significance of the effects of artesunate treatment and APP overexpression on dissociated neurons were tested using one-way ANOVA with Sidak’s multiple comparison test. Numeric values are given as mean ± standard error of the mean (SEM) or standard deviation (SD), as specified.

## Figures and Tables

**Figure 1 ijms-24-04623-f001:**
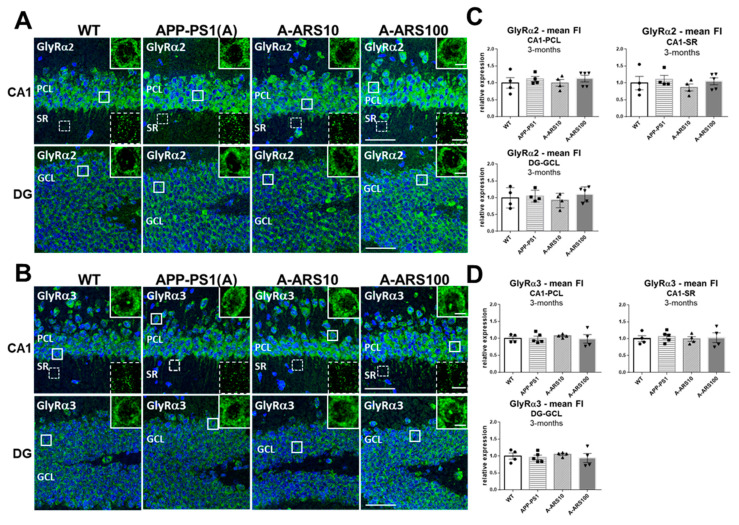
Level and regional distribution of GlyRα2 and GlyRα3 immunoreactivities in hippocampus sections of 3-month-old WT and APP/PS1 mice. (**A**,**B**) Representative IF images from brain cryosections of WT, APP/PS1 and APP/PS1 mice treated with two different doses (10 mg/kg and 100 mg/kg) of artesunate (A-ARS10, A-ARS100), labeled with subtype-specific antibodies for GlyRα2 (**A**) and GlyRα3 (**B**). Cell nuclei were revealed by 4′,6-diamidino-2-phenylindole (DAPI) staining. Similar staining patterns and regional distribution of GlyRα2 (**A**) and GlyRα3 (**B**) immunoreactivities can be seen in the CA1 region and the dentate gyrus (DG) of the hippocampus of young WT and APP/PS1 mice, as well as APP/PS1 mice treated with ARS. Both GlyR subtypes are enriched in the pyramidal cell layer (PCL) of CA1 and the granule cell layer (GCL) of DG. Only rather weak GlyRα2 (**A**) and GlyRα3 (**B**) immunostaining can be seen in the stratum radiatum (SR) of CA1. Insets show magnifications of the boxed windows in the PCL and SR of the CA1 region and GCL of the dentate gyrus, respectively. They show intense punctate GlyRα2 (**A**) and GlyRα3 (**B**) staining at the soma of CA1 pyramidal cells, DG granule cells and GlyR-positive clusters of different sizes within the SR of the CA1 region, respectively. Confocal maximum intensity projections (8 optical sections, 3.5-µm thick z-stack; insets showing cells—2 optical sections, 1-µm thick z-stack); scale bars: 50 µm; insets: 10 µm. (**C**,**D**) Quantification of GlyRα2 (**C**) and GlyRα3 (**D**) immunoreactivities in the PCL and SR of the CA1 region and GCL of the dentate gyrus of WT and APP/PS1 mice. Three randomly chosen fields within CA1 and two in the DG of each hippocampus were analyzed. *n* = 4–5 animals/group; means ± SEM of mean fluorescence intensities (FI).

**Figure 2 ijms-24-04623-f002:**
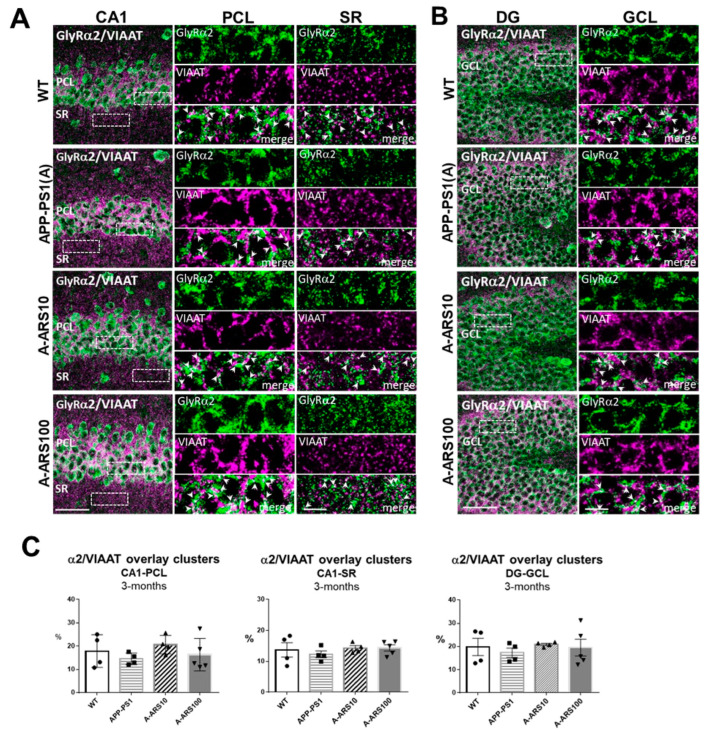
Subcellular localization of GlyRα2 clusters in hippocampus sections of 3-month-old WT and APP/PS1 mice. (**A**,**B**) Representative double channel IF images of the CA1 region (**A**) and the dentate gyrus (DG) (**B**) of WT, APP/PS1 and APP/PS1 mice treated with two different doses (10 mg/kg and 100 mg/kg) of artesunate (A-ARS10, A-ARS100). Brain cryosections were double labeled with an anti-GlyRα2 (green) and an anti-VIAAT (magenta) antibody. Right rectangular panels are magnifications of the boxed windows within the pyramidal cell layer (PCL) and the stratum radiatum (SR) of the CA1 region and granule cell layer (GCL) of DG, respectively, showing single channel acquisitions (GlyRα2, VIAAT) and merged images. Arrows in merged images point to overlapping clusters, indicating putative synaptic localized GlyRα2 clusters. Confocal maximum intensity projections (8 optical sections, 3.5-µm thick z-stack; rectangular panels—2 optical sections, 1-µm thick z-stack); scale bars: 50 µm; panels: 10 µm. (**C**) Quantitative analyses of overlapping clusters in hippocampus areas. Note the higher density of GlyRα2 positive/VIAAT positive clusters in the PCL of low-dose artesunate-treated APP/PS1 mice in comparison to untreated ones. Three randomly chosen fields within PCL and SR of CA1 and two in the dentate gyrus (DG) of each hippocampus were analyzed (*n* = 4–5 brains/group). The numbers of overlapping clusters were normalized to the total number of clusters positive for the GlyRα2 subunit and given as a percent (%). Means ± SEM.

**Figure 3 ijms-24-04623-f003:**
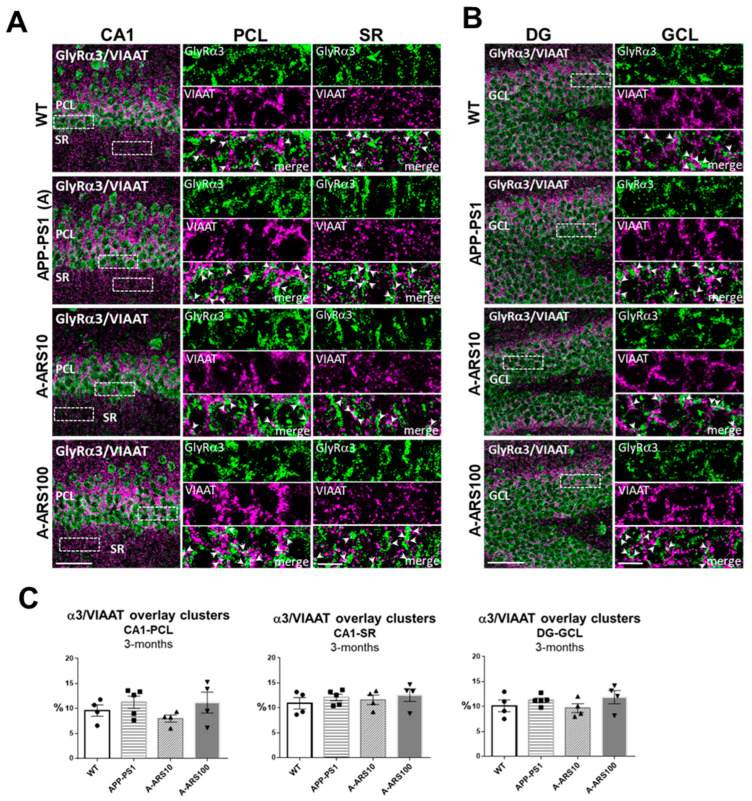
Subcellular localization of GlyRα3 clusters in hippocampus sections of 3-month-old WT and APP/PS1 mice. (**A**,**B**) Representative double channel IF images of the CA1 region (**A**) and the dentate gyrus (DG) (**B**) of WT, APP/PS1 and APP/PS1 mice treated with two different doses (10 mg/kg and 100 mg/kg) of artesunate (A-ARS10, A-ARS100). Brain cryosections were double labeled with an anti-GlyRα3 (green) and an anti-VIAAT (magenta) antibody. Right rectangular panels are magnifications of the boxed windows within the pyramidal cell layer (PCL) and the stratum radiatum (SR) of the CA1 region and granule cell layer (GCL) of DG, respectively, showing single channel acquisitions (GlyRα3, VIAAT) and merged images. Arrows in merged images point to overlapping puncta, indicating putative synaptic localized GlyRα3 clusters. Confocal maximum intensity projections (8 optical sections, 3.5-µm thick z-stack; rectangular panels—2 optical sections, 1-µm thick z-stack); Scale bars: 50 µm; panels: 10 µm. (**C**) Quantitative analyses of overlay clusters in hippocampus areas. Note the lower density of GlyRα3 positive/VIAAT positive clusters in the PCL of low-dose artesunate-treated APP/PS1 mice in comparison to untreated ones. Three randomly chosen fields within PCL and SR of CA1 and two in the dentate gyrus (DG) of each hippocampus were analyzed (*n* = 4–5 brains/group). The numbers of overlapping clusters were normalized to the total number of clusters positive for GlyRα3 subunit and given as a percent (%). Means ± SEM.

**Figure 4 ijms-24-04623-f004:**
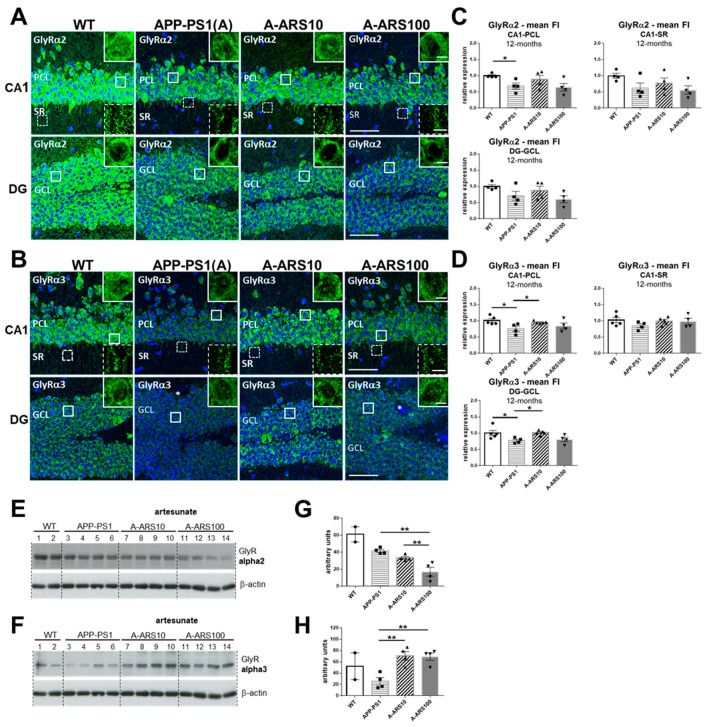
Level and regional distribution of GlyRα2 and GlyRα3 immunoreactivities in hippocampus sections of 12-month-old WT and APP/PS1 mice. (**A**,**B**) Representative IF images from brain cryosections of WT, APP/PS1 and APP/PS1 mice treated with two different doses (10 mg/kg and 100 mg/kg) of artesunate (A-ARS10, A-ARS100), labeled with subtype specific antibodies for GlyRα2 (**A**) and GlyRα3 (**B**). Insets show magnifications of the boxed windows in the pyramidal cell layer (PCL) and the stratum radiatum (SR) of the CA1 region and granule cell layer (GCL) of the dentate gyrus (DG), respectively. Representative micrographs show reduced GlyRα2 (**A**) and GlyRα3 (**B**) immunoreactivity within the CA1 region and DG of the hippocampus of 12-month-old APP/PS1 mice compared to WT. Comparing the rather intense GlyRα2 (**A**) and GlyRα3 (**B**) immunoreactivities in the somata of neurons of pyramidal (PCL) and granular cell layers (GCL), the decrease in the labeling intensity in the APP/PS1 transgene is evident. A discrete increase in the staining intensity of GlyRα2 (**A**) can be observed in the CA1 and DG of the hippocampus from APP/PS1 mice treated with low-dose (A-ARS10), but not with high-dose artesunate (A-ARS100) in comparison to the non-treated animals (APP/PS1). GlyRα3 immunoreactivity (**B**) is obviously increased in the hippocampus sections of all artesunate-treated APP/PS1 mice. The increase in staining intensity is more obvious in the hippocampus of low-dose artesunate-treated mice (A-ARS10). Note the well-preserved morphology of the nuclei, evidenced by DAPI in all sections. Confocal maximum intensity projections (8 optical sections, 3.5-µm thick z-stack; insets showing cells—2 optical sections, 1-µm thick z-stack); scale bars: 50 µm; insets: 10 µm. (**C**,**D**) Quantification of GlyRα2 (**C**) and GlyRα3 (**D**) immunoreactivities in the PCL and SR of the CA1 region and GCL of the dentate gyrus of WT and APP/PS1 mice. Note the significant decrease of both GlyRα2 (**C**) and GlyRα3 (**D**) immunoreactivity in cell soma-rich layers of the hippocampus from 12-month-old APP/PS1 mice in comparison to WTs. Only GlyRα3 immunoreactivity levels (**D**) are totally restored after treatment with artesunate, specifically at low-dose (A-ARS10). Asterisks in (**D**) indicate amyloid plaques. Three randomly chosen fields within CA1 and two in the DG of each hippocampus were analyzed. *n* = 4–5 animals/group; means ± SEM of mean fluorescence intensities (FI). Student’s *t*-test, * *p* < 0.05. (**E**,**F**) Representative Western blots probed with specific GlyRα2 and GlyRα3 antibodies, respectively, using hippocampal protein extracts from 12-month-old WT, untreated APP/PS1 and APP/PS1 mice treated with 10- or 100 mg/kg artesunate (A-ARS10; A-ARS100). (**G**,**H**) Quantification of mean grey intensities from Western blots are shown in (**E**,**F**) for GlyR subtypes. WTs were excluded from statistical analyses. Note the decrease of total protein levels of both GlyRα2 (**G**) and GlyRα3 (**H**) in untreated APP/PS1 mice. Artesunate restores protein levels of GlyRα3 (**F**,**H**) but not of GlyRα2 (**E**,**G**). Mean ± SEM. One-way ANOVA with Bonferroni’s multiple comparison tests. * *p* < 0.05, ** *p* < 0.01.

**Figure 5 ijms-24-04623-f005:**
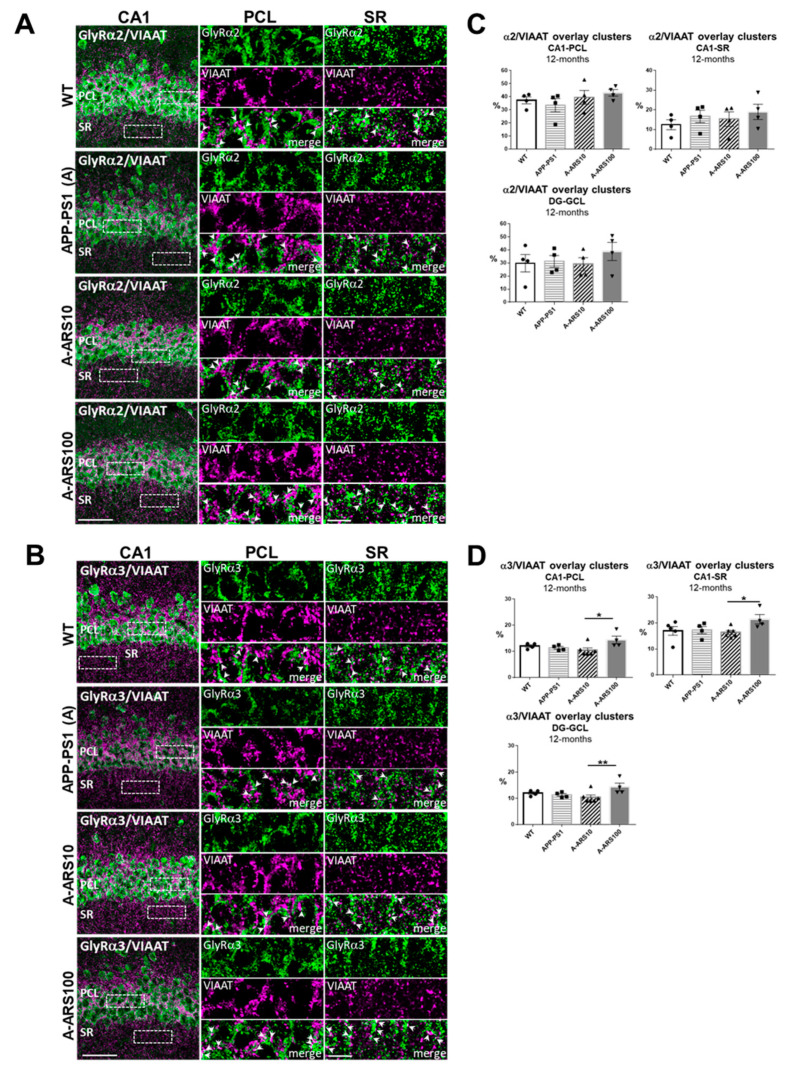
Subcellular localization of GlyRα2 and GlyRα3 clusters in hippocampus sections of 12-month-old WT and APP/PS1 mice. (**A**,**B**) Representative double channel IF images of the CA1 region of WT, APP/PS1 and APP/PS1 mice treated with two different doses (10 mg/kg and 100 mg/kg) of artesunate (A-ARS10, A-ARS100). Brain cryosections were double labeled with an anti-GlyRα2 (**A**) or anti-GlyRα3 (**B**) (green) and an anti-VIAAT (magenta) antibody. Right rectangular panels are magnifications of the boxed windows within the pyramidal cell layer (PCL) and the stratum radiatum (SR) of the CA1 region, respectively, showing single channel acquisitions (GlyRα2 (**A**), GlyRα3 (**B**), VIAAT) and merged images. Arrows in merged images point to overlapping puncta, indicating putative synaptic localized GlyRα2 and GlyRα3 clusters. Confocal maximum intensity projections (8 optical sections, 3.5-µm thick z-stack; rectangular panels—2 optical sections, 1-µm thick z-stack); scale bars: 50 µm; panels: 10 µm. (**C**,**D**) Quantitative analyses of GlyRα2 (**C**) and GlyRα3 (**D**) subunits overlapping VIAAT immunoreactivities in hippocampus areas of 12-month-old WT and APP/PS1 mice. Slices were double labeled with an anti-GlyRα2 or anti-GlyRα3 and an anti-VIAAT antibody. The numbers of overlapping GlyR subtype- and VIAAT-positive clusters were determined in the pyramidal cell layer (PCL) and the stratum radiatum (SR) of the CA1 region and granule cell layer (GCL) of the dentate gyrus (DG). Three randomly chosen fields within PCL and SR of CA1 and two fields in the dentate gyrus (DG) of each hippocampus were analyzed (*n* = 4–6 brains/group). The results represent the percentage of GlyRα2 (**A**) and GlyRα3 (**B**) facing VIAAT-positive clusters from the total number of respective GlyRα subtypes counted. Means ± SEM. One-way ANOVA with Bonferroni’s multiple comparison tests. * *p* < 0.05, ** *p* < 0.01.

**Figure 6 ijms-24-04623-f006:**
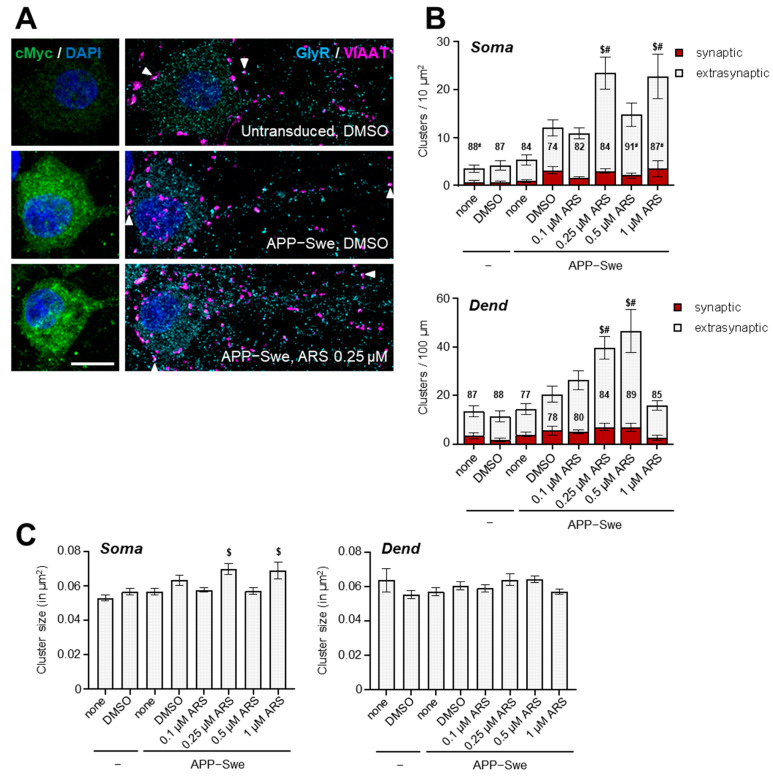
The effects of APP-Swe overexpression and artesunate treatment on GlyR clusters in dissociated hippocampal neurons. (**A**) Representative IF images of transduced hippocampal neurons of selected groups. cMyc-tagged APP-Swe transduced cells were manually selected for analysis based on cMyc immunofluorescence. Regions of interest encompassing either the entire soma or the proximal dendrite were selected and analyzed. The presence of synaptic (overlapping with presynaptic VIAAT signal) and extrasynaptic GlyR clusters (no overlap) can be clearly seen, as well as diffuse cytoplasmatic staining. (**B**) Quantification of the density of GlyR clusters at both the somatic (upper graph) and proximal dendritic (lower graph) compartments. Clusters were subdivided in putative synaptic (red bars) and extrasynaptic (light grey bars). GlyR clusters based on their overlap with the VIAAT signal. The numbers represent the percentage of extrasynaptic clusters from total clusters counted. Note a statistically significant increase in extrasynaptic cluster density for 0.25 and 1 µM artesunate for the somatic compartment, and for 0.25 and 0.5 µM artesunate for the dendritic compartment. Additionally, the percentage of extrasynaptic clusters was statistically significantly increased for the same artesunate dosages in the somatic compartment. (**C**) Quantitative analysis of cluster size. Note a statistically significant increase in cluster size for the 0.25 and 1 µM artesunate treatments in the somatic compartment only. Results are depicted as mean ± SEM. *n* = cells (22–24 per group). ^#^ *p* < 0.05 compared to APP-Swe-DMSO, ^$^ *p* < 0.05 compared to untransduced-DMSO (one-way ANOVA with Sidak’s multiple comparison tests).

## Data Availability

Details regarding data supporting reported results are available from the corresponding author, upon reasonable request.

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
