# Peer review of "Loss of Extrasynaptic Inhibitory Glycine Receptors in the Hippocampus of an AD Mouse Model Is Restored by Treatment with Artesunate"

_ijms, 2023, doi:10.3390/ijms24054623_

Round 1

Reviewer 1 Report

The authors of “The loss of extrasynaptic inhibitory glycine receptors in the 2 hippocampus of an AD-mouse model is restored by treatment 3 with artesunate” Have investigated an animal model of AD for the glycine receptor alpha 2 and 3 abundance and subcellular localization. This was investigated in multiple Hippocampal subregions at the ages of 3 and 12 month of w.t. animals, AD model animals with and without treatment with low or high dose of artesunate. They have found little alteration at 3 months, but a reduced abundance at 12 months for the AD model mouse compared with w.t., this reduction was rescued by low dose of artesunate.

While the study adds to our knowledge of glycine receptors under these conditions, I find the claims to be unsupported by the experiments performed due to a few reasons.

First, the antibodies used are not well validated. I assume (as it is not specified) that the alpha 2 antibody is GTX105634 and alpha3 is G0420. They are both not validated by knock-out (K.O.) and as such it is not known if they have unspecific binding to other targets in situ. While this might seem like a high bar, requiring K.O. validation for each antibody, the conclusions of this entire study rest on the unproven specificity of 2 antibodies.

Second, the colocalization of VGAT (VIITA) and a post synaptic receptor are not evidence of either synaptic or extra synaptic localization. VGAT would fill the buttons of presynaptic axons as it is associated with vesicles. This is not evidence of a release capable active zone adjacent to a postsynaptic scaffold.

Third the resolution needed to make that determination is lacking in conventional light microscopy. Especially in the Z direction and if analysis was done after z projecting the data as overlapping pixels might just mean unresolved z depth differences. In addition, it is unclear that all puncta in these images are coming from a membrane bound receptors. This adds to the confusion of total number of puncta to consider our 100% in the calculation of fraction synaptic receptors.

I would suggest adding Bassoon and Gephyrin staining if you want to know if these are close to release capable inhibitory synapses. And maybe doing Expansion Microscopy to improve Z resolution. However, without K.O. validated antibodies[1], it is hard to see how investing more effort would yield stronger evidence to these conclusions.

Other issues:

1.      Please report all cat# of antibodies used (better to use RRIDs).

2.      Line 54: not sure why spinal cord distribution of these receptors is needed in the paper.

3.      Background (neuropil) intensity seems to vary across conditions in the IHC. For example, Figure 2A, last column as well as Figure 4A first column both seems with much higher intensity all around. Were all conditions imaged under the same conditions, and are the images share the same LUT?

[1] A quick search of https://app.benchsci.com didn’t produce any results for validated antibodies.

Author Response

Reply to Reviewer 1

We thank Reviewer 1 for the constructive comments and remarks.

  1. “…, the antibodies used are not well validated. I assume (as it is not specified) that the alpha 2 antibody is GTX105634 and alpha3 is G0420. They are both not validated by knock-out (K.O.) and as such it is not known if they have unspecific binding to other targets in situ. While this might seem like a high bar, requiring K.O. validation for each antibody, the conclusions of this entire study rest on the unproven specificity of 2 antibodies.”

We see and understand the concerns of the Reviewer. We agree, that the K.O. validation of the GlyR alpha2 and alpha 3 antibodies used in our study would strongly support their specificity. Unfortunately, we do not possess glycine receptor KO animals and the generation and establishment of GlyR alpha2 and alpha 3 KO cells is a time consuming process and not manageable within the time limits of the revision period (10 days). However, we would like to note that both antibodies have been used successfully and published by others. The anti-alpha 2 GlyR antibody was used for Western Blot experiments by Zhu H et al., Nature. 2021; 599: 513-517, and for immunohistochemistry (IHC, IHC-fr) by Puri J, et al.  in J Cell Physiol. 2011; 226: 3169-3180. The anti-GlyR alpha3 antibody was used successfully for immunohistochemistry in the following publication: Acuña MA, et al. J Clin Invest. 2016; 126: 2547-2560. Now, we included these references also into the manuscript.

  1. “…, the colocalization of VGAT (VIITA) and a post synaptic receptor are not evidence of either synaptic or extra synaptic localization. VGAT would fill the buttons of presynaptic axons as it is associated with vesicles. This is not evidence of a release capable active zone adjacent to a postsynaptic scaffold.”

We agree, that quantifying the apparent overlap between VGAT (VIAAT) and a GlyR subtype has its limitations in detecting glycinergic synapses, as does – less or more – this approach independent of the pre- and postsynaptic marker used when analyzed by confocal microscopy. However, VGAT is considered a reliable marker of synaptic vesicles throughout the whole inhibitory presynaptic bouton (Chaudhry et al., J Neurosci. 1998;18:9733-50). To our knowledge VGAT is a well-accepted and widely used presynaptic marker for inhibitory GABAergic and inhibitory synapses in IHC and ICC examinations and was used in previous studies to dissect “synaptic” and “extrasynaptic” localized GlysR subunits in different brain regions, including the hippocampus (Aroeira, R.I et al, 2011, J Neurochem. 2011, 118, 339-353, Weltzien, F et al, J Comp Neurol. 2012, 520, 3962-3981), which prompted us to use the same marker for eventual correlations.

  1. “…the resolution needed to make that determination is lacking in conventional light microscopy. Especially in the Z direction and if analysis was done after z projecting the data as overlapping pixels might just mean unresolved z depth differences. In addition, it is unclear that all puncta in these images are coming from a membrane bound receptors. This adds to the confusion of total number of puncta to consider our 100% in the calculation of fraction synaptic receptors.”

We agree that the use of light microscopy imply certain limitations concerning the subcellular resolution and determining apposed localization of pre- and postsynaptic proteins in neurons, in particular analyzing brain-slices and maximal intensity projections by confocal microscopy, and thus to clearly detect and exact precisely quantify the synaptic localized GlyR subtypes. Considering these grounded remarks of the Reviewer in the new version of the manuscript we avoid the use of the term “synaptic localized” and replaced it with “overlapping with VIAAT”. Further, when presenting and discussing respective data in the results and discussion part of the manuscript, we are pointing to the limitations due to the method when interpreting data. We would like to notice, that using the calculation of fraction of “synaptic” receptors as percent we followed the method used in earlier publications (Aroeira, R.I et al, 2011, J Neurochem. 2011, 118, 339-353, Weltzien, F et al, J Comp Neurol. 2012, 520, 3962-3981).

In addition, we would like to underscore, that the main findings of our study are provided by comparative analyses of different experimental groups kept and processed under exactly similar experimental conditions and analyzed by the same methodology. Thus, in our view the results of this study demonstrating for the first time that 1) in the APP-PS1 mouse model of AD the protein levels of GlyR subunits are reduced in comparison to WTs and 2) this reduction in protein levels are restored to about WT level upon low-dose artesunate treatment and 3) the overall increased immunoreactivities in the hippocampus of artesunate treated mice are provided by immunoreactive clusters that are not overlapping with presynaptic VIAAT immunoreactivities - are not substantially affected by the limitations of the methodology. Furthermore, the findings in brain sections are supported by the results of cell culture experiments which allow a more precise analysis of quantitative differences in the numbers of overlapping VIAAT and GlyR immunoreactivities.

  1. “I would suggest adding Bassoon and Gephyrin staining if you want to know if these are close to release capable inhibitory synapses. And maybe doing Expansion Microscopy to improve Z resolution. However, without K.O. validated antibodies[1][1], it is hard to see how investing more effort would yield stronger evidence to these conclusions.”

Thank you for this remark. We agree, that staining with Bassoon and Gephyrin could provide additional data and support to the findings of our study. However, these would imply one or two new series of staining which by the number of experimental groups would imply a quite high number of new sections and for given groups even new experimental animals and further imaging sessions, which cannot be done in a foreseeable period of time. In addition, the analysis by confocal microscopy would raise the same questions when analyzing data, and at the moment we do not have access to Expansion Microscopy. Please see also answer to point 1.

Other issues

“Please report all cat# of antibodies used (better to use RRIDs).”

The cat. nr. of the used antibodies has been included.

“Line 54: not sure why spinal cord distribution of these receptors is needed in the paper.”

By including in the introduction part of the manuscript some data referring to the distribution of GlyRs also in the spinal cord our purpose was to further underscore the heterogeneity in their temporal and regional distribution in the CNS.

“Background (neuropil) intensity seems to vary across conditions in the IHC. For example, Figure 2A, last column as well as Figure 4A first column both seems with much higher intensity all around. Were all conditions imaged under the same conditions, and are the images share the same LUT?”

Sections from untreated and artesunate-treated mice for each time point were labelled simultaneously using the same batches of solutions to avoid differences through immunolabeling conditions. Laser power and all imaging settings were identical for all samples stained in one session. All settings, including laser power, were kept constant between all recordings in one data set.

Our manuscript has been checked by a native English-speaking colleague and the changes are marked in the revised version of the manuscript.

Reviewer 2 Report

The authors of "The loss of extrasynaptic inhibitory glycine receptors in the hippocampus of an AD-mouse model is restored by treatment with artesunate" presents the effects of different artesunate treatments on an Alzheimer disease model at differentt time points, at 3 and 12months.

The introduction is well reasoned and it continues the work of the last years of the laboratory. M&M are well described and argued. Since they don't know the therapeutical mechanim of action of artesunate, they come up with several hipothesis about its therapeutic effect and its impact on the GlyR at different stages. But from my point of view the article, and what it os worse the procedure, has to main flaws: they didn't include groups of WT animals treated with artesunate and incorrect statistical analisis.

I think it is really necessary to include groups of WT animal treated with the same dose of active principle, especially since it is not known the action mechanism. We don't know if the changes observed in the treated APPPS1 animals will also appear on WT animals. And therefore, it is reflecting just the effect of artesunate or could have a therapeuric effect due to its impact on the accumulation of beta amyloid.

As I said the statistical analysis is not correct, except on the analysis done to the neuronal culture. t-test is an analysis only valid when there are only 2 groups, that it is not the case. Best case scenario it should be an ANOVA wit a post-hoc test to compare between groups. But if they would include treated WT animals, then the appropiate t4est would be a 2-way ANOVA, considering genotype and treatment, to observe their effect and their interaction.

Some other concerns.

Fluorescence images: I know it is a max projection, but the images seem to be saturated... So, not surprising that many groups have really similar values of intensity. I don`t know if it is the best method to study it.

Results of figure 2 & 3: They decided to show them as a percentage, that it is a great mode to normalize. But is there any difference in the raw numbers? Is there a difference number of clusters between the groups?

Figure 3: The western blots only show 2 WT animals. I know that it is due to gels had 15 lanes, and they prioritized the APPPS1 animals, but there should be at least 3 animals, just 2 invalidates any further statistical analysis, and if there is any problem, like in the case of figure F, you end up with a huge error. Looking at the gel in the supplementary figures it seems that there was a transfer problem affecting the lower left part of the membrane.

Figure 5: I see no reason for not showing the immunostaining

Figure 6: On C, So, there is an effect of artesunate on the cluster size at 0,25 and 1 microM, but there is not effect at 0,5. For me it is hard to understand it. I would understand the the effect disapears or changes at higher concentrations, but going and coming back... Hard to figure it out.

I don't have Supplementary figure 1 that it is cited in the text.

A minor typo, but the text is full of double spaces between some words...

Author Response

Reply to Reviewer 2

We thank Reviewer 2 for the constructive comments and remarks.

  1. “… I think it is really necessary to include groups of WT animal treated with the same dose of active principle, especially since it is not known the action mechanism. We don't know if the changes observed in the treated APPPS1 animals will also appear on WT animals. And therefore, it is reflecting just the effect of artesunate or could have a therapeuric effect due to its impact on the accumulation of beta amyloid.”

We agree, that studying the effects of artemisinins under normal physiological conditions is sure a scientific relevant issue and might help to understand their mechanisms of action also under pathological conditions, such as Alzheimer’s disease. However, the main purpose of our earlier and present studies was and is to dissect changes that appear at the level of inhibitory synapses (GABAergic and glycinergic) in the AD-brain, more specifically in the hippocampus and to test whether these AD-relevant changes are in some way modulated by the seemingly multipotent artemisinins. Thus, our experimental design for these studies does not include groups of WT mice treated with artemisinins. The long-term nature of our animal experiments make not possible the recruitment and evaluation of new experimental animals in a foreseeable period of time. Further studies have to investigate and show whether the observed changes of inhibitory synapses in artemisinins treated APP-PS1 mice are due through direct effects on the inhibitory synapses, or indirect effects through e.g. antiamyloidogenic or anti-inflammatory effects or even independent of the pathomechanisms of AD, as the Reviewer suggested.

  1. “…, As I said the statistical analysis is not correct, except on the analysis done to the neuronal culture. t-test is an analysis only valid when there are only 2 groups, that it is not the case. Best case scenario it should be an ANOVA wit a post-hoc test to compare between groups.”

We considered the critique of Reviewer 2 concerning the choice of statistical tests for calculating statistical significance for quantitative data of the immunohistochemistry and western blot experiments. We reanalyzed all our data using one- way ANOVA with Bonferroni multiple comparison test to appreciate statistical significance between groups. Corresponding changes have been done in the manuscript text and figures. 

  1. “…I know it is a max projection, but the images seem to be saturated... So, not surprising that many groups have really similar values of intensity. I don`t know if it is the best method to study it"

We are aware about the limitations of light microscopy concerning quantifications. However for a comparative analyses of sections from mice from different time points, different experimental groups and of different hippocampal subregions we had to use the same laser power and identical imaging settings, which resulted in some sections to “saturation”. No other methods are available to analyze differences in protein levels and in subcellular protein localization in different hippocampal subregions.

  1. "Results of figure 2 & 3: They decided to show them as a percentage, that it is a great mode to normalize. But is there any difference in the raw numbers? Is there a difference number of clusters between the groups?"

Yes, there are clearly differences also in numbers between groups looking on the raw data, however these values show a higher variability within the groups than normalized values. 

  1. “Figure 3: The western blots only show 2 WT animals. I know that it is due to gels had 15 lanes, and they prioritized the APPPS1 animals, but there should be at least 3 animals, just 2 invalidates any further statistical analysis, and if there is any problem,   like in the case of figure F, you end up with a huge error. Looking at the gel in the   supplementary figures it seems that there was a transfer problem affecting the lower left part of the membrane..”

Thank you for this remark. We agree, that for statistical analyses at least three animals are necessary. The two lanes of WT animal protein extracts were not included in the statistical analyses and are shown only as reference. The statistical analyses of WBs includes only the extracts of APP-PS1 animals, without and with artesunate. This is now specified also in the Figure legend. 

  1. “Figure 5: I see no reason for not showing the immunostaining”

Immunostaining images have been included in Figure 5.

  1. "Figure 6: On C, So, there is an effect of artesunate on the cluster size at 0,25 and 1 microM, but there is not effect at 0,5. For me it is hard to understand it. I would understand the the effect disapears or changes at higher concentrations, but going and coming back... Hard to figure it out."

We agree. We do not have a clear explanation for this finding. However, this is only a minor aspect of this analysis, which, in our view, does not affect basically the “message” of this experiment.

Other issues

“I don't have Supplementary figure 1 that it is cited in the text.”

Supplemental figure 1 has been uploaded together with the manuscript.

“A minor typo, but the text is full of double spaces between some words...”

We checked the manuscript and we hope we have corrected all these typing errors. In addition, our manuscript has been checked by a native English-speaking colleague and the changes are marked in the revised version of the manuscript.

Reviewer 3 Report

Authors analyzed protein levels of GABAergic synapses in the hippocampus of APP-PS1 mice and also treatment with low doses of Artesunate improved localization of subunits of GlyR of hippocampus in APP-PS1 mice. In previous article (kiss etal., 2021, Artesunate restores the levels of inhibitory synapse proteins and reduces amyloid-B and CTFs of the APP in AD mouse model ), authors showed that low doses of artimisine stabilized GABAergic inhibitory synapses and improvesdabeta plaques . Experiments designed well and results supported their hypothesis. 

Author Response

Reply to Reviewer 3

We thank Reviewer 3 for his positive comments and appreciations to our manuscript.

Our manuscript has been checked by a native English-speaking colleague and the changes are marked in the revised version of the manuscript. 

Reviewer 4 Report

This is a very well written paper. Some of the changes cited are small, but others are quite significant, and the treatment does appear to restore the function. This is an interesting area in Alzheimer's research.

Author Response

Reply to Reviewer 4

We thank Reviewer 4 for his positive comments and appreciations to our manuscript.

Our manuscript has been checked by a native English-speaking colleague and the changes are marked in the revised version of the manuscript. 

Round 2

Reviewer 1 Report

I find that the revisions done based on my comments are not sufficient for publication in IJMS. The title of the paper as well as most figures still refer to synaptic vs. extra synaptic receptors that the authors acknowledge they cannot determine. The fact a previous study used the antibodies, doesn’t validate them. The fact a previous study claimed synaptic localization by overlap with VGAT is similarly invalid.

Author Response

Reply to Reviewer 1

“…The title of the paper as well as most figures still refer to synaptic vs. extra synaptic receptors that the authors acknowledge they cannot determine. The fact a previous study used the antibodies, doesn’t validate them. The fact a previous study claimed synaptic localization by overlap with VGAT is similarly invalid.”

In the new version of the manuscript we performed further corrections in figures legend and manuscript text replacing the term “synaptic” localization with “putative synaptic” localization. In addition, we added at the end of the discussion part of the manuscript a paragraph summarizing the potential limitations of our methodological approach and further outlook how these could be overcome by using higher resolution microscopic techniques and KO-validation of the antibodies – as suggested by the Editor. Please see also reply to Editor.

Reviewer 2 Report

I'm happy to see that the authors have corrected the statistical analysis of their experiments together with many changes in the text and figures. 

However, the major criticisms continue being present in the paper. The effect of artemisinin on physiological conditions is necessary to understand it in pathological conditions. 

I understand that to compare different animals, regions, etc, you have to keep the same laser power and imaging settings. But if some images are saturated, it means that the settings were not properly stablished. Also, it exists different techniques allowing to overcome light microscopy limitations, from Electron Microscopy to Super-resolution techniques on Light Microscopy.

So, I don't know the journal policy to answer, of course 10 day period doesn't allow to make any major changes in the article as it was asked. So, I say to the journal to reconsider prolonging this period.

Author Response

Reply to Reviewer 2

“…However, the major criticisms continue being present in the paper. The effect of artemisinin on physiological conditions is necessary to understand it in pathological conditions. 

I understand that to compare different animals, regions, etc, you have to keep the same laser power and imaging settings. But if some images are saturated, it means that the settings were not properly stablished. Also, it exists different techniques allowing to overcome light microscopy limitations, from Electron Microscopy to Super-resolution techniques on Light Microscopy.

So, I don't know the journal policy to answer, of course 10 day period doesn't allow to make any major changes in the article as it was asked…”

Thank you Reviewer 2 for his understanding, that in the condition of a timely limited revision period new experiments involving inclusion of WT animals in our experimental design cannot be managed. However, we added now at the end of the discussion part of the manuscript a paragraph summarizing the potential limitations of our methodological approach and further outlook how these could be overcome by using higher resolution microscopic techniques and KO-validation of the antibodies – as suggested by the Editor- as well as the potential usefulness of experiments with artemisinins in WT animals, to better understand the mechanisms of action of these substances. Please see also reply to Editor.

Round 3

Reviewer 2 Report

Sorry for the delay, but it has been a hard decision to make. I feel that the main criticisms are still present and they would need time to be solved. Otherwise, science world is demanding to publish continuously, and this rhythm is not good to make good science. From my point of view the experiment has an inadequate design. Is this invalidating the work done after? I done think so, but it affects the conclusions. 

Since the authors tuned down their conclusions, accepting the limitations of them, I say to publish in the present form, even if I think that it should be improved.